# Biomarkers of Affective Dysregulation Associated with In Utero Exposure to EtOH

**DOI:** 10.3390/cells13010002

**Published:** 2023-12-19

**Authors:** Nune Darbinian, Nana Merabova, Gabriel Tatevosian, Mary Morrison, Armine Darbinyan, Huaqing Zhao, Laura Goetzl, Michael Edgar Selzer

**Affiliations:** 1Center for Neural Repair and Rehabilitation (Shriners Hospitals Pediatric Research Center), Lewis Katz School of Medicine, Temple University, Philadelphia, PA 19140, USA; nmerabova@gmail.com (N.M.); dr.tatevosian@gmail.com (G.T.); 2Medical College of Wisconsin-Prevea Health, Green Bay, WI 54304, USA; 3Center for Substance Abuse Research, Lewis Katz School of Medicine, Temple University, Philadelphia, PA 19140, USA; mary.morrison@tuhs.temple.edu; 4Department of Psychiatry, Lewis Katz School of Medicine, Temple University, Philadelphia, PA 19140, USA; 5Department of Pathology, Yale University School of Medicine, New Haven, CT 06520, USA; armine.darbinyan@yale.edu; 6Center for Biostatistics and Epidemiology, Department of Biomedical Education and Data Science, Lewis Katz School of Medicine, Temple University, Philadelphia, PA 19140, USA; tuf08292@temple.edu; 7Department of Obstetrics & Gynecology, University of Texas, Houston, TX 77030, USA; laura.goetzl@uth.tmc.edu; 8Department of Neurology, Lewis Katz School of Medicine, Temple University, Philadelphia, PA 19140, USA

**Keywords:** FASD, depression, biomarkers, brain development, exosomes, serotonin receptors, dopamine receptors, FAS

## Abstract

Introduction: Children with fetal alcohol spectrum disorders (FASD) exhibit behavioral and affective dysregulation, including hyperactivity and depression. The mechanisms are not known, but they could conceivably be due to postnatal social or environmental factors. However, we postulate that, more likely, the affective dysregulation is associated with the effects of EtOH exposure on the development of fetal serotonergic (5-HT) and/or dopaminergic (DA) pathways, i.e., pathways that in postnatal life are believed to regulate mood. Many women who use alcohol (ethanol, EtOH) during pregnancy suffer from depression and take selective serotonin reuptake inhibitors (SSRIs), which might influence these monoaminergic pathways in the fetus. Alternatively, monoaminergic pathway abnormalities might reflect a direct effect of EtOH on the fetal brain. To distinguish between these possibilities, we measured their expressions in fetal brains and in fetal brain-derived exosomes (FB-Es) isolated from the mothers’ blood. We hypothesized that maternal use of EtOH and/or SSRIs during pregnancy would be associated with impaired fetal neural development, detectable as abnormal levels of monoaminergic and apoptotic biomarkers in FB-Es. Methods: Fetal brain tissues and maternal blood were collected at 9–23 weeks of pregnancy. EtOH groups were compared with unexposed controls matched for gestational age (GA). The expression of 84 genes associated with the DA and 5-HT pathways was analyzed by quantitative reverse transcription polymerase chain reaction (qRT-PCR) on microarrays. FB-Es also were assayed for serotonin transporter protein (SERT) and brain-derived neurotrophic factor (BDNF) by enzyme-linked immunosorbent assay (ELISA). Results: Six EtOH-exposed human fetal brain samples were compared to SSRI- or polydrug-exposed samples and to unexposed controls. EtOH exposure was associated with significant upregulation of DA receptor D3 and 5-HT receptor HTR2C, while HTR3A was downregulated. Monoamine oxidase A (MAOA), MAOB, the serine/threonine kinase AKT3, and caspase-3 were upregulated, while mitogen-activated protein kinase 1 (MAPK1) and AKT2 were downregulated. ETOH was associated with significant upregulation of the DA transporter gene, while SERT was downregulated. There were significant correlations between EtOH exposure and (a) caspase-3 activation, (b) reduced SERT protein levels, and (c) reduced BDNF levels. SSRI exposure independently increased caspase-3 activity and downregulated SERT and BDNF. Early exposure to EtOH and SSRI together was associated synergistically with a significant upregulation of caspase-3 and a significant downregulation of SERT and BDNF. Reduced SERT and BDNF levels were strongly correlated with a reduction in eye diameter, a somatic manifestation of FASD. Conclusions: Maternal use of EtOH and SSRI during pregnancy each was associated with changes in fetal brain monoamine pathways, consistent with potential mechanisms for the affective dysregulation associated with FASD.

## 1. Introduction

Prenatal exposure to ethanol (EtOH; PE) causes variable combinations of somatic, facial, cognitive, and behavioral abnormalities, known collectively as “fetal alcohol spectrum disorders” (FASD), the most severe of which is called “fetal alcohol syndrome” (FAS). FASD is the leading cause of preventable cognitive disabilities in the US [1,2,3,4], with a prevalence of 2–5% [5,6,7,8]. Of the estimated 80,000 children born with FASD each year in the US, half go undiagnosed. Children with FASD have small stature, characteristic facial abnormalities, and prominent cognitive and behavioral deficits, including symptoms of affective dysregulation, such as hyperactivity, irritability, anxiety, and depression [9,10,11,12]. Light or moderate EtOH use by 14-year-old children who had been exposed to EtOH prenatally was also associated with long-term negative behavioral and developmental outcomes [13,14]. Children with higher levels of PE exhibited higher levels of negative affect at 1 year and reported more depressive symptoms at 6 years [15,16]. The mechanisms for these behavioral and affective abnormalities are not known, but in other contexts, synaptic monoaminergic transmission pathways are implicated in mood regulation [9,17,18,19,20,21,22,23].

**Depression and drinking in pregnancy**. Depressed women drink more alcohol than nondepressed women [24], and conversely, PE is a major risk factor for mood disorders [25]. More than 20% of pregnant women have elevated depressive symptomatology, and 4.7% of pregnant women report treatment with serotonin-specific reuptake inhibitors (SSRIs; [26,27,28,29,30,31]). Thus, EtOH use is generally associated with an increased incidence of depression [32,33,34], and this is also true during pregnancy [26,35]. 

**Involvement of the 5-HT and DA pathways in affective dysregulation**. There is a very large body of evidence linking the monoamines 5-HT, DA, and norepinephrine (NE) to the regulation of affect. Although a comprehensive review of this literature is beyond the scope of this paper, the most effective treatments for both depression and anxiety disorders have been pharmacological agents that increase brain levels of one or more of these neurotransmitters. This includes monoamine oxidase inhibitors, which block the breakdown of all three monoamines after they have been released by presynaptic terminals in the brain [36], SSRIs, which inhibit the reuptake of 5-HT after it has been released by presynaptic terminals [37], and even serotonergic psychedelics, which bind to 5-HT receptors [38]. Of relevance, SSRIs have been effective in the treatment of generalized anxiety disorder, the most common affective disorder in children [39]. Pharmacological depletion of each of the three monoamine synaptic transmitters activates depression, but this seems not to be true for subjects who have not suffered from or have a family history of major depression [40]. Thus, it has been argued that monoamines may play a role in mood disorders, but only if there is a genetic predisposition. Clearly, the regulation of mood is a complicated process and a recent meta-umbrella review of the literature concluded that evidence for the serotonin theory of depression is lacking [41]. The methodology of that review has been challenged [42], and in the search for molecular markers to anticipate the emergence of affective dysregulation in FASD, it is reasonable to start with the monoaminergic pathways, especially those relating to 5-HT and DA activity. 

**Abnormalities in monoaminergic pathways are found in FASD.** SSRIs may potentiate EtOH-related neurotoxicity and possibly contribute to a 10-fold increased risk of FASD [43]. While SSRIs may induce some fetal malformations in humans [44], this may be due to the association between SSRI use and increased EtOH intake [45]. In rodents, 5-HT agonists prevented alcohol-related apoptosis of neurons and glia in brain regions that contain 5-HT cells [46,47], possibly by increasing levels of anti-apoptotic proteins [48], and impaired the development of the 5-HT system, including significant epigenetic modifications [49,50] and altered transcriptional regulation [51]. Changes in the activity of the serotonin transporter (SERT) in platelets were associated with both alcoholism and clinical depression in patients with hepatitis B infection [52], although the mechanism of the altered activity is not fully known. Neuronal adaptive cell responses to SSRIs include genetic and epigenetic regulation of the expression and activity of SERT, its post-translational modifications and specific interactions with a set of diverse partners, and its trafficking to and away from the plasma membrane [53]. In addition to 5-HT, SERT transports several other substances to the fetus from maternal blood [54,55]. Thus, if maternal EtOH use alters SERT activity in the placenta or in the fetus, this could alter fetal exposure to several drugs, which might have significant neurodevelopmental implications. 

In experimental animals, the 5-HT and DA pathways are affected by PE in the same directions as they are affected in childhood FASD, depression, attention deficit hyperactive disorder (ADHD), oppositional defiance disorder (ODD), and autism spectrum disorders (ASD) [10,21,56,57,58], although much is still unknown. This overlap in biochemical pathways parallels the behavioral symptom overlap and suggests that many cases of FASD go undiagnosed [59]. In rhesus monkeys, moderate PE contributed to neurodevelopmental impairments and disrupted several neurotransmitter systems, including monoaminergic pathways, altering the SERT gene polymorphic region, and disrupting levels of 5-HT and DA metabolites in cerebrospinal fluid (CSF). Thus, abnormal monoaminergic pathways might underlie some of the psychiatric disorders reported in FASD [60]. It is not known whether the brains of fetuses whose mothers use EtOH during pregnancy show such abnormalities in monoaminergic pathways, and if so, whether these are due to the direct effects of EtOH on fetal brain cells or an indirect effect due to the frequent use of SSRIs in women who use alcohol. In the present study, we wish to determine whether prenatal exposures to EtOH and SSRIs have synergistic or opposite effects on 5-HT and DA systems, sometimes exerting opposite effects via multiple receptors and downstream genes. 

Currently, fetal brain tissue examination is not routinely possible in human pregnancies. We previously showed that fetal brain-derived exosomes (FB-Es) isolated non-invasively from the maternal blood contained molecular markers that could be used diagnostically and that some of these markers were associated with anatomical abnormalities characteristic of FAS [61,62,63,64]. In the present study, we have analyzed fetal brains and the cargos of FB-Es isolated from the blood of pregnant women who drank EtOH during pregnancy for abnormalities of 5-HT and DA pathways predictive of affective dysregulation to determine whether the results from fetal brains and FB-Es were congruent, and whether any such abnormalities are associated with the direct effects of EtOH on the fetal brain, or are indirect, due to the concurrent use of SSRIs. The results might lead to the development of prenatal tests to predict which at-risk fetuses are likely to suffer from the behavioral abnormalities associated with FASD. 

## 2. Methods

### 2.1. Clinical Samples

A comparison was performed between pregnant women who consumed EtOH [63,64,65], those who consumed SSRIs, those who consumed more than one drug (polydrug; Table 1), and individually matched control pregnant women who did not use illicit drugs or take CNS-active medications. The selection of cases and controls was based on the availability of intact fetal brain tissues, matching maternal blood samples, and the availability of data for matching sex and GA [61,62,63,64,65,66]. Thus, each EtOH- and/or drug-exposed fetus involved in a comparison was individually paired with a GA- and sex-matched control. Consenting mothers were enrolled between 9- and 23-weeks GA (Tables in [63,64,65]) under a protocol approved by our Institutional Review Board (IRB). All assays were performed in triplicate. Data from both sexes were combined.

Assessment of EtOH Exposure in Pregnancy. Maternal EtOH, SSRI, and other drug exposures were determined with a face-to-face questionnaire that also included questions regarding many types of drugs/medications used [61,62,63,64,65,66,67]. Exposure status was based on self-reported EtOH use (modified timeline followback). Women were first screened for EtOH use, and then tissue was collected. Other substances (opioids, SSRIs, and amphetamines) were measured for previous studies in placenta transporter activities.

Depression status was ascertained using the Center for Epidemiologic Studies Depression Scale—Revised (CESD-R) survey with a cutoff score of 16 (CESD-R ≥ 16) [68,69]. EtOH exposure was defined as current daily use, and samples were matched based on the last incidence of EtOH consumption, as indicated by the survey. EtOH dose was calculated as the total number of drinks consumed in a week multiplied by the number of weeks of exposure. EtOH exposure was assessed using a detailed questionnaire based on measures adapted from the NICHD PASS study [70]. EtOH consumption for each week since conception (2 weeks after the last menstrual period) was self-reported using visual/photographic guides of different types of drinks to estimate the actual EtOH dose. Women admitting to any EtOH use were classified as EtOH exposed (see all details in [63]. 

Subject Recruitment. Women reporting EtOH use (or no EtOH use) since conception were grouped in GA windows: 9–23 weeks gestation (1st–late 2nd trimester). GA was determined by a dating ultrasound performed immediately prior to recruitment; at the GAs examined, ultrasound can accurately determine GA ± 10 days [71]. We collected fetal brain and matching maternal blood samples from 20 women who used EtOH, 20 using multiple other drugs, and 20 healthy non-drug-using controls. 

Tissue collection. Fetal brain tissue from subjects undergoing elective termination of pregnancy was collected according to an IRB-approved protocol. Surgical tissue samples were collected immediately by a trained study coordinator; both fresh and snap-frozen samples were transferred to the laboratory within 40–60 min. Then aliquots were either used for RNA and protein extraction or kept in liquid nitrogen for future use. The whole forebrain was used in this study. Initial histologic staining of brain tissues from the Biobank confirmed that we had collected mostly cerebral cortex [63,65]. Alcohol-exposed or control brain samples were used previously (as we indicated in [63,65] to study oligodendrocyte (OL) and neuronal markers. Other drug-exposed cases (SSRI, amphetamine, and opioids) are unique for this study, although matching maternal blood was studied for exosomal synaptic biomarkers and miRNAs in [61,62,66,72], and more new manuscripts are being prepared for publications with opioid- or alcohol-exposed brain tissues and matching maternal blood to study placental transporters or mtDNA damage. Initial histologic staining of brain tissues from the Biobank confirmed that we had collected mostly cerebral cortex.

### 2.2. RNA Preparation

Total RNA was isolated from fetal brain and brain-derived exosomes using the RNeasy Kit (Qiagen, Valencia, CA, USA) with on-column DNA digestion. 

Gender determination was carried out using SuperScript One-Step RT-PCR with Platinum Taq (Life Technologies, Carlsbad, CA, USA), BioRad C1000 Touch Thermal Cycler, and gender-determining SRY primers. 

Real-time qRT-PCR. The qRT-PCR reaction was performed using the One-Step FAST SYBR Green PCR Master Mix (Qiagen). For relative quantification, the expression level of genes was normalized to the housekeeping gene β-actin. 

qRT-PCR Assays. Human dopamine and serotonin pathway array (Cat. # 330231 PAHS-158ZR, Qiagen, Valencia, CA, USA) for 84 genes was assayed by real-time PCR with the Applied Biosciences Cycler using fetal brain RNAs, primers, and Cyber Green mix (Qiagen Universal PCR Master Mix). Real-time PCR experiments were performed on an Applied Biosystems instrument with the following thermal-cycling procedure: 95 °C for 10 min, followed by 40 cycles of 95 °C for 15 s, and 56 °C for 1 min, as specified in the RNA assay protocol provided by the Applied Biosystems.

RT^2^ Profiler™ PCR Array Human Dopamine and Serotonin Pathway (cat# Qiagen, Valencia, CA, USA). Expression of 86 genes regulating the dopamine and serotonin pathways was assayed for: 

Neurotransmitter Receptors: Dopamine Receptors DRD1, DRD2, DRD3, DRD4, DRD5.

Serotonin (5-hydroxytryptamine) receptors: HTR1A, HTR1B, HTR1D, HTR1E, HTR1F, HTR2A, HTR2B, HTR2C, HTR3A, HTR3B, HTR4, HTR5A, HTR6, and HTR7.

Neurotransmitter Synthesis and Degradation: Dopamine metabolism COMT, DBH, DDC, MAOA, and TH.

Serotonin Metabolism MAOA, MAOB, TDO2, TPH1, and TPH2.

Dopamine and Serotonin Transporters: SLC6A3 (DAT) and SLC6A4 (SERT).

Signal Transduction: cAMP and Protein Kinase A Signaling ADCY1, ADCY2, ADCY3, ADCY5, CASP3, CDK5, CREB1, DUSP1 (PTPN16), FOS, MAPK1 (ERK2), PPP1R1B (DARPP32), and PRKACA.

AKT and PI3 Kinase Signaling AKT1, AKT2, AKT3, GSK3A, GSK3B, PIK3CA (p110-alpha), and PIK3CG.

Phospholipase A2 Pathway ALOX12, CYP2D6, PDE10A, PDE4A, PDE4B, PDE4C (QuantiNova Symbol: AC008397.2), PDE4D, and PLA2G5.

Phospholipase C SignalingITPR1, PLCB1, PLCB2, PLCB3.

G-Protein Coupled Receptor Regulation ADRB1, ADRB2, APP, ARRB1, ARRB2, GRK2, GRK3, GRK4, GRK5, GRK6, SNCA, and SNCAIP.

Dopamine and Serotonin Gene Targets: BDNF, EPHB1, GDNF, GFAP, NR4A1 (NUR77), NR4A3 (NOR1), PDYN, PTGS2 (COX2), SLC18A1 (VMAT1), SLC18A2 (VMAT2), and SYN2.

### 2.3. Droplet Digital PCR (ddPCR)

For absolute quantitation of mRNA copies, ddPCR was performed using the QX200 ddPCR system. Fifty nanograms of human fetal total RNA was used with the 1st Strand cDNA Synthesis Kit (Qiagen, Valencia, CA, USA). After reverse transcription, the cDNA (300 dilution) aliquots were added to the BioRad master mix to conduct ddPCR (EvaGreen ddPCR Supermix, BioRad, Hercules, CA, USA). The prepared ddPCR master mix for each sample (20-μL aliquots) was used for droplet formation. PCR conditions: Activation 95 °C 5 min, PCR 45 cycles at 95 °C 10 s, 60 °C 20 s, 72 °C 30 s, melting curve (95–65 °C), cool to 40 °C 30 s. The absolute quantity of DNA per sample (copies/µL) was calculated using QuantaSoft Analysis Pro Software (AP) (Bio-Rad, Hercules, CA, USA) to analyze ddPCR data for technical errors (Poisson errors) [63,65]. The Poisson distribution relates the probability of a given number of events occurring independently in a sample when the average rate of occurrence is known and very low. Accurate Poisson analysis requires optimizing the ratio of the number of positive events (positive droplets) to the total number of independent events (the total number of droplets). A greater total number of droplets results in higher accuracy. With 20,000 droplets, the above ddPCR protocol yields a linear dynamic range of detection between 1 and 100,000 target mRNA copies/µL. The estimated error is negligible compared with other error sources, e.g., pipetting, sample processing, and biological variation. The ddPCR data were exported to Microsoft Excel (Microsoft 365) for further statistical analysis. 

Primers (IDT Inc., Coralville, IA, USA).

β-actin: S: 5′-CTACAATGAGCTGCG TGTGGC-3′, 

AS: 5′-CAGGTCCAGACGCAGGATGGC-3′, 

BDNF: S: 5′-CAGGGGCATAGACAAAAG-3′, AS: 5′-CTTCCCCTTTTAATGGTC-3′,

SERT (HTTLPR): S: 5′- ATGCCAGCACCTAACCCCTAATG-3′,

AS: 5′-GAGGGACTGAGCTGGACAACCCAC-3′

SRY: S: 5′-CAT GAA CGC ATT CAT CGT GTG GTC-3′; AS: 5′-CTG CGG GAA GCA AAC TGC AAT TCT T-3′.

### 2.4. Flow Cytometry

Plasma samples were analyzed according to the previously published protocols with modifications [63,65,73]. In brief, cells were washed with a cold phosphate-buffered saline (PBS) cocktail with 0.1% BSA and 1% protease inhibitors (Sigma, St. Louis, MO, USA). Cells were passed through a 70 μM mesh, and 10,000 cells were placed onto 96-well plates and incubated with fluorescein isothiocyanate (FITC)-conjugated primary antibody for 1 h. βIII tubulin was used as a neuronal marker in the developing CNS, and GFAP was used as a marker for astrocytic staining. Proportions were quantified using 5000 cells and GUAVA FACS (fluorescence-activated cell sorting) software (5.3.1) [63,65]. 

### 2.5. ELISA Quantification of Exosomal Proteins

SERT, BDNF, synaptic proteins, and CD81 (American Research Products-Cusabio) were quantified by human-specific ELISAs according to suppliers’ directions. Levels of synaptic proteins were quantified by human ELISA kits for synaptophysin (American Research Products/CSB-E17406H, Waltham, MA, USA), synaptotagmin (BioMatik-EKU07545, Wilmington, DE, USA), synaptopodin (BioMatik-EKU07541, Wilmington, DE, USA), and neurogranin (American Research Products/CEA404HU, Waltham, MA, USA). ELISA data was statistically evaluated by Excel and statistical analysis tools (CurveExpert for ELISA statistics (CUSABIO) or APP 96-well Plate Assay Data Analysis Software 5.0.apk (Cloud-Clone, Katy, TX, USA), available online.

### 2.6. Synaptosome Extract Preparation

Synaptic extracts were prepared from frozen brain tissues using the Syn-Per Extraction Kit (Reagent #87793, Thermo, Waltham, MA, USA). 

### 2.7. Quantitative Western Blots

Fifty micrograms of proteins in Laemmli sample buffer were heated at 95 °C for 10 min and separated by 10% sodium dodecyl sulfate-polyacrylamide gel electrophoresis (SDS-PAGE), then transferred to supported nitrocellulose membranes. Primary antibodies: anti-SERT rabbit polyclonal (AB10514P, Millipore, Burlington, MA, USA), anti-synapsin, and anti-α-tubulin clone B512 (Sigma-Aldrich, St. Louis, MO, USA). Secondary IRDye^®^ antibodies were used to detect band intensity (normalized to tubulin) using the Odyssey^®^ CLx Imaging System and LiCor dyes.

Antibodies. Neuronal class III β-tubulin (TUJ1) monoclonal antibody (Alexa Fluor-labeled) from Covance Inc. (Berkeley, CA, USA), anti-cleaved caspase-3 from Sigma, Alexa Fluor^®^ labeled from Covance. Anti-GFAP from Millipore.

### 2.8. Caspase Glo-3/7 Assay

Caspase-3 activity was assayed as a marker for apoptosis in synaptosomal and cytosomal lysates, prepared using Syn-Per Synaptic Protein Extraction Reagent (Thermo Scientific, Waltham, MA, USA). Caspase-3 activity was measured with the Caspase-Glo™ 3/7 Assay Kit, using the substrate DEVD-aminoluciferin (Promega, Madison, WI, USA; [61,74,75,76,77]. Luminescence was recorded as RLU/sec on a Turner Designs Luminometer TD-20/20 (Promega).

### 2.9. Isolation of Fetal Brain-Derived Exosomes (FB-Es) from Maternal Plasma, and ELISA Quantification of Exosomal Proteins

Human FB-Es were isolated as described previously [61,62,63,72]. Two hundred and fifty microliters of plasma was incubated with 100 mL of thromboplastin-D (Fisher Scientific, Inc., Hanover Park, IL, USA) and cocktails of protease and phosphatase inhibitors. After centrifugation, supernatants were incubated with exosome precipitation solution (EXOQ; System Biosciences, Inc., Mountainview, CA, USA), the resultant suspensions were centrifuged at 1500× *g* for 30 min at 4 °C, and pellets resuspended in 400 mL of distilled water with protease and phosphatase inhibitor cocktail for immunochemical enrichment of exosomes. To isolate exosomes from fetal neural sources, total exosome suspensions were incubated for 90 min at 20 °C with 50 μL of 3% bovine serum albumin (BSA) (Thermo Scientific, Inc., Waltham, MA, USA) containing 2 μg of mouse monoclonal IgG1 antihuman contactin-2/TAG1 antibody (clone 372913, R&D Systems, Inc., Minneapolis, MN, USA) or MBP antibody that had been biotinylated (EZLink sulfo-NHS-biotin System, Thermo Scientific, Inc., USA). Then, 10 μL of Streptavidin-Plus UltraLink resin (PierceThermo Scientific Inc., Waltham, MA, USA) in 40 μL of 3% BSA was added, and the incubation continued for 60 min at 20 °C. After centrifugation at 300× *g* for 10 min at 4 °C and removal of supernatants, pellets were resuspended in 75 μL of 0.05 mol/L glycine-HCl (pH 3.0), incubated at 4 °C for 10 min, and recentrifuged at 4000× *g* for 10 min at 4 °C. Each supernatant was mixed in a new 1.5 mL Eppendorf tube with 5 mL of 1 mol/L Tris-HCl (pH 8.0) and 20 μL of 3% BSA, followed by addition of 0.4 mL of mammalian protein extraction reagent (M-PER; Thermo Scientific Inc., Waltham, MA, USA) containing protease and phosphatase inhibitors prior to storage at −80 °C. For exosome counts, immunoprecipitated pellets were resuspended in 0.25 mL of 0.05 mol/L glycine-HCl (pH 3.0) at 4 °C with a pH of 7.0 and 1 mol/L Tris-HCl (pH 8.6). Exosome suspensions were diluted 1:200 to permit counting in the range of 1–5 × 10^8^/mL, with an NS500 nanoparticle tracking system (NanoSight, Amesbury, UK). 

### 2.10. Statistical Analysis

Analysis was performed using IBM SPSS Statistics for Windows, Version 25.0, released in 2017 (Armonk, NY, USA), as described previously [63,65]. All data are represented as the mean ± standard error for all performed repetitions. Means were analyzed by one-way ANOVA, with Bonferroni correction for multiple comparisons. Statistical significance was defined as *p* < 0.05. Sample numbers are indicated in the figure legends. Data from ddPCR, which measures absolute quantities of DNA per sample (copies/µL), were processed using QuantaSoft Analysis Pro Software (Bio-Rad) to analyze for technical errors (Poisson errors), and then exported to EXCEL for further statistical analysis.

### 2.11. Ethics: Human Subjects

Consenting mothers were enrolled at between 9 and 23 weeks’ gestation, under a protocol approved by our Institutional Review Board (IRB). This protocol involved no invasive procedures other than routine care. Maternal EtOH exposure was determined with a face-to-face questionnaire that also included questions regarding many types of drugs/medications used [61,62,63,65,66]. The questionnaire was adapted from that designed to identify and quantify maternal EtOH exposure in the NIH/NIAAA Prenatal Alcohol and SIDS and Stillbirth (PASS) study [70]. 

All procedures involving the collection and processing of blood and tissues were done according to NIH guidelines through a trained study coordinator. All investigators were trained annually to complete Citi Program—Human Subject Training, Biohazard Waste Safety Training and Blood-Borne Pathogens Training, and all other required training.

Eligibility Criteria. The blood and brain samples were obtained according to NIH guidelines through a trained study coordinator. Samples were collected regardless of sex, ethnicity, and race.

Treatment Plan. Each patient was asked to sign a separate consent form for research on blood and tissue samples. The blood obtained was processed for the collection of serum and plasma. No invasive procedures were performed on the mother other than those used in her routine medical care. Fetal tissues were processed for RNA or protein isolation. 

Risk and Benefits. There were very small risks of loss of privacy, as with any research study in which protected health information is viewed. The samples were depersonalized before they were sent to the lab for analysis. There were no additional risks of blood sampling, as this was only performed in subjects with clinically indicated venous access. There was little anticipated risk from obtaining 2–3 cc of blood, but a well-trained study coordinator collected all samples.

There was no direct benefit to the research subjects from participation, but there is significant potential benefit for future FAS subjects and the general population. This research represents a reasonable opportunity to further the understanding, prevention, or alleviation of a serious problem affecting the health or welfare of FAS patients.

Informed Consent. Consent forms were maintained by the study coordinator and were not sent to the investigator with the samples. The de-identified log sheets and IRB protocol were sent by the study coordinator to the principal investigator with each blood and tissue sample. This sheet contained an assigned accession number as well as the age, sex, ethnicity, and race of the patient. Except for the accession number, no identification was kept on the blood and tissue samples.

## 3. Results

Subjects were recruited into EtOH-, SSRI-, amphetamine-, and polydrug-using groups and compared with a non-using control group (Table 1). 5-HT and DA pathways in fetal brain and FB-Es were surveyed by mRNA arrays (Figure 1) and other advanced and sensitive methods, including isolation of exosomes from maternal blood, tissue samples from fetuses, RNA arrays for mRNA screening, ELISA for protein determinations, qRT-PR, and ddPCR for quantification of RNAs of interest. Microarrays were employed to study the effects of EtOH and SSRI exposure on the mRNA expression of genes for proteins that are in the pathway toward synthesis of 5-HT (MAOA, MAOB), or in the pathways to remove 5-HT (SERT, MAO), or in the pathway to synthesize SERT or to inactivate or downregulate SERT, including SSRIs (the genes shown in Figure 1, on the left). A similar approach was used for DA and its receptors (Figure 1, on the right). 

### 3.1. Prenatal EtOH, SSRI and Polydrug Exposures Are Associated with Disruption of 5-HT and DA Pathways

We performed a series of RNA array assays, using real-time PCR and ddPCR, to study genes regulating the 5-HT and DA pathways and confirmed mRNA data by protein studies (ELISA, flow cytometry, and quantitative Western blot assay) to correlate changes in mRNA expression with protein levels. Cases with detailed maternal histories of depression, prescription medications, SSRIs, multiple drugs, and EtOH use (Table 1) were used to study genes and proteins regulating 5-HT and DA pathways (Figure 1), including 5-HT and DA receptors, transporters, and downstream target genes. Maternal EtOH use and maternal depression during pregnancy were each associated with increased neurotoxicity.

The literature and our data provide strong evidence for the key role of synaptic proteins and their upstream molecules, including 5-HT-Rs, in the mechanism of EtOH-mediated neurotoxicity. Here, we show that prenatal EtOH exposure was associated with altered levels of 5-HT-Rs and SERT in the fetal brain and at the human fetal synapse and that the degree of neurotoxicity is directly correlated with the alterations in SERT. In utero EtOH exposure and/or maternal depression were associated with alterations in expression and activity of the 5-HT/DA pathways, receptors (5-HT-R group; D3, D4), or transporter groups (SERT, DAT). Expression of 84 genes was analyzed by qPCR array (Figure 1) in the presence of EtOH and other drugs (Table 1) using RNA from fetal brain tissues. EtOH was associated with downregulation of 5-HT receptor HTR3A (1.1-fold decrease for HTR3A, Figure 2A) and upregulation of HTR2C mRNA (4-fold, Figure 2A), suggesting the possible existence of compensatory mechanisms to maintain the overall levels of 5-HT receptors after EtOH exposure. Downregulation of mRNA for the 5-HT receptor HTR3A was also found in SSRI-exposed fetal brains (12-fold decrease, Figure 2A). EtOH exposure was associated with upregulation of DA receptors DRD3 and DRD4 in the fetal brain (2- to 6-fold increase), while SSRI and polydrug use downregulated DRD3 in the fetal brain (Figure 2B). We also studied the effect of EtOH and polydrug use on the synthesis and degradation of 5-HT, by assaying MAOA and MAOB, and on the synthesis or degradation of DA, by assaying MAOA in the fetal brain (Figure 2C). EtOH upregulated both MAOA and MAOB (30- and 12-fold increase, respectively) but downregulated MAPK1 mRNA (18-fold decrease; Figure 2C) in the downstream signal transduction pathway (cAMP/PKA pathway). Further, EtOH exposure increased DAT mRNA expression (9-fold; Figure 2D) but reduced (11-fold) SERT mRNA levels in the fetal brain (Figure 2D). Combined EtOH and SSRI exposure inhibited mRNA expression of BDNF, which is downstream of the 5-HT and DA pathways, much more effectively than either alone (2-fold decrease by EtOH, 1.3-fold decrease by SSRIs, and 14-fold decrease by both). Downregulation of mRNA expression of BDNF and upregulation of caspase-3 mRNA (4-fold increase), another target gene in the cAMP/PKA pathway, in EtOH, SSRI, and polydrug-exposed fetal brains (Figure 2E). The RNA array also included immediate-early genes (IEGs), BDNF and GDNF, downstream of SERT and DAT in the 5-HT and DA pathways. 

RNA expression of PI3K/AKT pathway target genes AKT2 and AKT3 in EtOH, SSRI, and polydrug-exposed fetal brains is presented in Figure 2F. Again, while EtOH exposure downregulated AKT2 mRNA expression (9-fold decrease), it upregulated AKT3 (6-fold, Figure 2F). Finally, a series of target genes downstream of serotonin and dopamine transporters, such as synaptic genes and synapsin, were assayed in the fetal brain (Figure 2G). While SNCA and SYN2 were downregulated by EtOH exposure (28-fold and 2-fold decrease), SNCAP was upregulated (1.5-fold increase). SSRI and polydrug exposure inhibited the mRNA expression of GDNF and GFAP, which are downstream genes in the 5-HT/DA pathway. Downregulation for GFAP mRNA was greatest in the cases with SSRI exposure, although EtOH downregulated significantly too (55- and 9-fold decrease). While EtOH exposure increased mRNA expression of the monoaminergic pathway target gene GDNF and SSRI inhibited its expression, polydrug exposure, which could include EtOH and/or SSRIs, but not necessarily) showed the greatest effect, reducing GDNF expression 8-fold (Figure 2H). 

### 3.2. Effects of Early or Late EtOH Exposure and SSRI on Serotonin Transporter, SLC6A4, BDNF and Caspase-3 Gene Expression

Next, we used fetal brain samples to compare neuronal injury in EtOH-exposed cases vs. controls by measuring RNA expression for SERT, BDNF, and caspase-3 genes in a microarray. The influence of the trimester (1st or 2nd) and GA in the weeks in which EtOH exposure occurred (Figure 3). Early and late EtOH exposure produced different effects on the serotonin transporter SLC6A4 and on the genes regulating its downstream targets, caspase-3 and BDNF, in the brain (Figure 3A). Downregulation of serotonin transporter gene (2- to 4-fold decrease) and BDNF (2- to 18-fold decrease) and up-regulation of caspase-3 gene expression (4.1- to 8-fold increase) were found in brain samples exposed to EtOH during the 1st trimester, while slight increases in SERT and BDNF gene expressions or no effect were detected in samples from 2nd trimester EtOH exposures. The effects of combined exposure to EtOH and SSRI on mRNA expression of serotonin transporter SLC6A4 (24-fold decrease), caspase-3 (5-fold increase), and BDNF (12-fold decrease) were significantly greater than the effects of either one alone when exposure occurred during the first trimester (Figure 3B), but not during the 2nd trimester (Figure 3C). Late EtOH exposure caused less brain injury than early exposure (2-fold vs. 5-fold increases in caspase-3 expression, respectively). SSRI exposure during the 2nd trimester did not induce statistically significant increases in caspase-3 gene expression and did not significantly inhibit 5-HT and BDNF gene expression in the fetal brain (Figure 3C). 

### 3.3. Effects of Gestational Age, EtOH and SSRIs on Fetal Synaptosome SERT Protein Expression, and on Neuronal Injury

Exposure to both EtOH and SSRIs was associated with reduced SERT levels in fetal brain synaptosomes (Figure 4A). We assayed activated caspase-3 in the same cases (Figure 4B) to determine whether maternal EtOH use and/or maternal depression during pregnancy results in increased apoptosis at the early and late stages of CNS development. Caspase-3 activation was enhanced with EtOH exposure in brain synaptosomes from 20 fetuses (Figure 4C). Increased caspase-3 activity in fetal brain synaptosomes was detected in the EtOH group (Figure 4C; *p* = 0.03). Caspase-3 levels were higher in the synaptosome fraction than in the cytoplasm. Synaptic caspase-3 activity was positively correlated with GA in both control and EtOH-exposed brains (r = 0.73, *p* = 0.04, Figure 4D), as expected, since some activation of caspase-3 is part of normal development. 

### 3.4. Downregulation of SERT and Synaptic Proteins by EtOH Exposure Is Highly Correlated in Synaptosomes and Brain-Derived Exosomes

To extend our observations to the level of protein expression, we studied the effects of EtOH on synaptic markers by ELISA. 

To allow non-invasive monitoring of fetal brain molecular development, we have developed methods to isolate FB-Es from maternal blood [61,62,63,65,66,72], and demonstrated that purified FB-Es can be used to detect fetal CNS damage caused by in utero EtOH exposure. In the present study, EtOH exposure reduced levels of SERT and BDNF, as well as the synaptic proteins synaptophysin, REST, synaptopodin, and synapsin-2 in fetal brain synaptosomes and exosomes (n = 10 EtOH-exposed and 10 controls). (Figure 5), and the findings in synaptosomes correlated closely with those in exosomes, particularly for synapsin (Table 2). SERT levels in synaptosomes were assayed by ELISA (n = 6/group). EtOH and SSRI each reduced SERT levels in synaptosomes, and the effects of the combination of EtOH and SSRI were significantly greater than the effects of either exposure alone (Figure 5A). Exposure to EtOH and SSRI each also inhibited SERT levels in FB-Es, as assayed by ELISA (n = 6/group), although the combined effect was not much greater than the effect of SSRIs alone (Figure 5B). EtOH also reduced BDNF levels in synaptosomes (Figure 5C) and in FB-Es (Figure 5D; n = 10/group). Reduction in synapsin levels by EtOH exposure in synaptosomes (Figure 5E) and in exosomes (Figure 5F; n = 10) was demonstrated by quantitative Western blot.

### 3.5. EtOH- and SSRI-Exposed Fetal Brains Show Increased Neuronal Apoptosis

Previously, we used flow cytometry to demonstrate that EtOH exposure increased apoptosis in fetal neuronal cells [65]. Here, we used flow cytometry to measure caspase-3 activity in neurons and astrocytes exposed to EtOH and SSRIs. EtOH-induced increases occur in both cell types, but more strongly in neurons. Flow cytometry analysis of fetal neuronal cells exposed to EtOH and SSRI showed increased neuronal apoptosis in SSRI cases (Figure 6A vs. Figure 6B, circled in green) and even more in EtOH cases (Figure 6C). However, the combination of EtOH and SSRI produced far more apoptosis than either exposure alone (Figure 6D).

### 3.6. Increased Apoptosis in Astrocytes by SSRI Exposure

Flow cytometry of GFAP-positive fetal brain cells showed an increase in caspase-3 activity in SSRI-exposed fetal brain tissue compared to control fetal brain cells (Figure 7), indicating a toxic effect of prenatal SSRI exposure on astrocytes in the fetal brain. This was also seen with EtOH exposure [65].

### 3.7. Severe Downregulation of SERT by Prenatal EtOH and Maternal Depression

Because SSRI exposure reduced SERT protein levels in FB-Es, we used ddPCR and ELISA (n = 10/group) to determine whether maternal depression without SSRI treatment was also associated with a downregulation of SERT expression in FB-Es. Both maternal EtOH consumption and depression were associated with reduced SERT mRNA copy number (37,560 copies/μL in controls vs 29730 copies/μL for EtOH and 21,480 copies/mL for depression) (Figure 8A) and SERT protein (Figure 8B). EtOH and depression each reduced SERT protein levels in fetal brain exosomes (Figure 8B). The effects were greater for the combination of EtOH and depression than for either condition alone, although the difference in mRNA copy number between cases with depression alone and those with both EtOH + depression did not reach statistical significance. These results suggest that untreated maternal depression may itself affect the development of the serotonergic system in the brain and, when combined with exposure to EtOH, might intensify the affective dysregulation of FASD. 

### 3.8. Correlations between Human Fetal Eye Size and Expression of SERT and BDNF in FB-E

Because this study did not involve postnatal follow-up to determine which at-risk fetuses would have developed FASD, and certainly not affective dysregulation, we correlated some of the most prominent molecular abnormalities with fetal eye diameter, which we have shown previously is a convenient quantitative anatomical hallmark of FASD that can be correlated with molecular markers in the fetal brain [63]. There was a significant correlation between the reduction in eye size (difference between the EtOH-exposed fetus and its paired control) and the reduction in exosomal SERT (Figure 9A; R = 0.81) and BDNF (Figure 9B; R = 0.70) protein levels. Thus, it is possible that FB-E SERT levels, in particular, might be a molecular marker to predict the development of affective dysregulation in children who were exposed to EtOH in utero.

## 4. Discussion

FASD has been subdivided into several syndromes [12]. Its most severe form, FAS, includes the full spectrum of craniofacial, somatic, and neurobehavioral abnormalities (abnormal mood or impaired behavioral regulation, attention deficit disorder, impulse control, and cognitive impairment). Partial forms include partial fetal alcohol syndrome (pFAS), alcohol-related neurodevelopmental disorder (ARND), and alcohol-related birth defects (ARBD) [12]. In pFAS, some but not all the diagnostic features of FAS are present. ARBD represents physical abnormalities due to EtOH exposure, while in ARND, a broad range of neurodevelopmental/neurobehavioral problems are seen without anatomical features. Neurobehavioral abnormalities, including dysregulated mood, are seen in a majority of children with all of these subtypes except ARBD. It might not be surprising that children with unusual behavioral, somatic, cognitive, and/or facial features might find it difficult to fit in with their peers and might react to rejection by acting out or by demonstrating other evidence of mood dysregulation, with the abnormalities of monoaminergic molecular pathways that accompany it [78]. However, the present data demonstrate that such molecular abnormalities are already present prenatally. There might be several reasons for this. It could reflect a direct effect of EtOH on the developing brain, or it might be due to transplacental movement of molecules associated with maternal depression, which, as described in the introduction, is seen frequently in pregnant women who drink EtOH, or even the effects of SSRIs, with which depressed women are often treated. The data presented here suggest that maternal EtOH consumption, maternal depression, and maternal SSRI use, each independently and in combination, are associated with abnormalities at the mRNA and protein levels of molecular pathways (i.e., 5-HT and DA) that are associated with abnormal mood regulation in adult humans. The altered expression patterns are consistent with what is known about the affective disorders seen in FASD postnatally [79]. Moreover, the abnormalities correlate with evidence that the affected fetuses show anatomical abnormalities of FASD, i.e., small eyes or small brains, suggesting that had they survived, they might have met the criteria for FASD postnatally. In this study, among the EtOH-associated abnormalities detected by RNA arrays was a dramatic downregulation of SERT expression, an increase in DAT expression, a decrease in BDNF and synapsin mRNA expressions, and an increase in caspase-3 expression. In most cases, the effects of SSRIs were similar to those of EtOH, and the effects of combined exposure to EtOH + SSRIs were greater than those of either one alone. Most of the RNA changes were reflected by protein expression changes, most dramatically for inhibition of SERT and synaptic proteins and activation of caspase-3.

### 4.1. Neuronal Injury in FASD and Depression

Neuronal injury has been implicated in the pathogenesis of FASD [80,81,82]. The neurotoxic effects of EtOH in animals depend on GA and EtOH doses [83,84,85,86]. Several molecular mechanisms are involved in EtOH-induced neural injury in animals [87,88,89,90,91] and include induction of apoptosis (caspase-3 activation), defects in synaptic plasticity, and impaired 5-HT-dependent plasticity [92,93,94,95,96]. The mechanisms of EtOH neurotoxicity have not been well studied in humans, but we and others have shown that EtOH exposure is associated with the activation of apoptosis and cell damage [65,75,89]. 

### 4.2. Differential Associations of EtOH and SSRI Exposure on Changes in Gene Expression for 5-HT and DA Receptors and Pathways

There is now much evidence linking the monoaminergic systems with mood dysregulation on the one hand and with alcohol abuse on the other, but how the findings all fit is still not clear. The effects of EtOH and SSRI are similar, but their effects on the 5-HT pathway are different from their effects on the DA pathway. Previous data in primary cultures [75,97,98,99], fetal brain tissues, and FB-Es from maternal blood [61,62,63,65] demonstrated an association between EtOH exposure and several neuronal and OL molecular markers. The literature and the present data suggest that abnormal expression of the 5-HT and DA receptors (5-HT-Rs; D1, D2, β1, β2) and transporters (SERT, DAT) may play a role in the mechanism of EtOH-mediated neurotoxicity and potentially in the affective dysregulation of FASD. In the present study, SSRIs and EtOH had similar but independent effects on the 5-HT and DA pathways. In humans, the SERT gene encodes a solute carrier of family 6, member 4 (SLC6A4). Four genetic variants were found to be strongly related to low IQ at age 8, but only in the offspring of mothers who were moderate drinkers [100]. 5-HT2C receptor blockade reversed SSRI-associated basal ganglia dysfunction and potentiated the “antidepressant” effects of SSRIs in mice [101]. There is a functional significance to the differential response of 5-HT receptors, e.g., HTR3A vs. HTR2C [102]. In the mouse hippocampus, each HTR had a unique expression pattern and contributed differently to hippocampal functions, such as learning, memory, and affect. In the ventral hippocampus, HTR1A, 2A, 2C, and 7 play a role in mood- and anxiety-related behavior and are involved in the antidepressant effects of SSRIs. Expression of the 5-HT2C receptor in mice modulated alcohol intake, which may be relevant to human alcoholism [103]. Increased cortical expression of 5-HT2CR mRNA had previously been found in major depressive suicide victims [104]. HTR3B is associated with antisocial behavior in alcoholism [105]. In mice, PE from embryonic day 7 (E7) to E15–18 reduced the number of 5-HT neurons in several fetal brain regions [106] and caused abnormal transcription of HTR3A due to promoter DNA methylation [107]. The development of new antidepressant therapies based on the combination of SERT inhibition with different pharmacological activities on the 5-HT system [108]. The 5-HT system and several 5-HT receptors have also been associated with alcoholism, making this system a target for pharmacotherapy [109]. There is also some evidence suggesting that long-term antidepressant use reduces brain 5-HT concentrations [41]. 5-HT receptors are localized in brain areas involved in mood regulation (e.g., the hippocampus and prefrontal cortex) and in the regulation of neurotransmitter systems implicated in the pathophysiology of major depression (DA). 5-HT3 receptors may be a relevant target in the treatment of affective disorders [110]. There is an association between dopamine receptors D3 (DRD3) and D4 (DRD4) and affective dysregulation [111]. D4 interacts with adverse life events to predict maternal sensitivity via emotion regulation [112]. Recent studies demonstrate the important role of neurotransmitters, D1 and D2 dopamine receptors, and NMDA receptors in EtOH susceptibility [113,114]. PE induces functional and structural plasticity in D1 receptor-expressing neurons [115]. Data on the effects of PE, maternal depression, and SSRI use on 5-HT and DA pathway genes in the fetal brain is still limited, as until now there has been no data linking EtOH or SSRIs to abnormalities in the 5-HT system. 

Here, we demonstrate the differential association of EtOH or SSRI exposure with changes in 5-HT and DA pathway gene expression in the fetal brain. First, we demonstrated that in utero EtOH exposure and/or maternal depression were associated with altered expression and activity of the 5-HT and DA pathways, receptors (5-HT-R group; D3, D4), and transporter groups (SERT, DAT). EtOH was associated with the downregulation of the 5-HT receptor HTR3A but the upregulation of mRNA for another 5-HT receptor, HTR2C, indicating the possibility of compensatory mechanisms maintaining the levels of 5-HT receptors and specific targets for EtOH exposure. SSRI exposure was also associated with downregulated HTR3A mRNA levels. By contrast, EtOH exposure was also associated with the upregulation of DA receptors DRD3 and DRD4 in the fetal brain, while maternal SSRI and polydrug use was associated with the downregulation of DRD3 in the fetal brain. The data in the human fetal brain also demonstrated an association between EtOH exposure and increased levels of DA receptor protein. EtOH exposure was also associated with increased DAT mRNA expression but reduced SERT mRNA levels in the fetal brain, suggesting possible differential effects of EtOH exposure on DAT and SERT. 

### 4.3. Differential Effects of Early vs. Late Exposure to EtOH and SSRI on mRNA and Protein Expression for SERT, BDNF and Caspase-3

There is evidence for reductions in the levels of BDNF in a mouse model of FASD [9]. In the present study, EtOH and SSRI exposure were associated with statistically significant synergistic effects on mRNA expression of SLC6A4 (the gene for SERT), caspase-3, and BDNF only for early exposures. This is significant because many women may have drunk alcohol before they knew they were pregnant. Thus, early detection of biomarkers for FASD may be especially important. Activation of caspase-3 in the nucleus and cell body cytoplasm is a well-known marker of apoptosis. The association between exposure to EtOH and/or SSRI and increased caspase-3 activity in neuronal and astrocytic cells observed in the flow cytometry studies of the present study is consistent with the reduction in brain size and the cognitive impairments characteristic of many children with FAS and FASD [92,116,117].

Among the novel findings reported here is a synergistic effect in the association of EtOH and SSRI exposure with SERT downregulation and caspase-3 and BDNF upregulation in the fetal brain. The possible association of untreated maternal depression with abnormalities in human fetal development is also novel. The data have uncovered potential targets (SERT, BDNF) that might help to prenatally predict and perhaps even ameliorate FASD, or mood dysregulation. 

### 4.4. Increased Levels of Activated Caspase-3 in Synaptosomes

Transient non-apoptotic activation of caspases has been described in dendritic pruning during normal development [118,119] and some other cell functions. Local, non-apoptotic caspase-3 activation is involved in dendritic spine loss and synaptic dysfunction in Alzheimer’s disease and in the rapid loss of dendritic spines seen with synaptic long-term depression (LTD) in striatal projection neurons forming the indirect pathway. Systemic treatment with a caspase inhibitor prevented both the dendritic spine pruning and the physiological deficit of LTD without interfering with the ongoing dopaminergic degeneration (discussed in [120]). In the present study, the increased levels of activated caspase-3 demonstrated in synaptosomes from EtOH-exposed fetuses are consistent with the known loss of brain volume and the cognitive disabilities of children with FAS [121] and the loss of dendritic spines previously shown in animal fetuses exposed to EtOH [122,123] and in human FAS [116]. Whether this involves monoaminergic neurons remains to be determined.

### 4.5. Brain-Derived Exosomes

FB-Es were used in this study for the first time. Maternal plasma miRNAs have been used to predict infant outcomes and may be useful to classify difficult-to-diagnose FASD subpopulations [124,125,126]. We have developed methods to isolate fetal brain-derived CNS exosomes from maternal blood to demonstrate downregulation of neuronal and synaptic proteins, OL markers, and activation of the pro-apoptotic marker caspase-3 [61,62,66,72]. This strategy was further developed in the present study to investigate SERT, BDNF, and synaptic biomarkers of depression in FB-Es. The results suggested that exosomal BDNF, SERT, or synapsin from maternal serum may be a powerful tool to assess early depression and neurotoxicity as early as the first trimester.

### 4.6. Biomarkers

The current findings suggest that molecular markers (e.g., low levels of SERT, BDNF, and synapsin mRNA and protein, and high levels of caspase-3) in FB-Es, which can be isolated noninvasively from maternal blood, may prove valuable as predictors of FASD. SERT, synapsin, and BDNF mRNA and protein levels were reduced in the EtOH-exposed fetal brains, and the same pattern of downregulation was seen in FB-Es isolated from the plasma. Thus, we may be able to use these abnormalities to predict which fetuses with EtOH exposure will develop FASD and affective dysregulation and to institute therapies to prevent it. To our knowledge, the use of FB-Es to study the association between EtOH exposure and abnormalities of human fetal 5-HT and DA pathways has not been reported previously.

Both EtOH use and affective disorders in the mother might influence the fetal development of those monoaminergic pathways that have been implicated in affective dysregulation, i.e., the 5-HT, DA, and noradrenergic receptors, transporters, and downstream pathways. In the present study, not only EtOH exposure but also SSRIs and/or maternal depression were associated with abnormalities in fetal brain development. If so, then it may be important for physicians to carefully choose doses of SSRIs when treating depression in pregnant women. 

### 4.7. Prognostic Value of FB-E Monoaminergic Biomarkers

Of special interest are abnormalities in FB-E SERT and BDNF levels correlated with reductions in eye diameter, one of the most constant facial features of FASD. This suggests that these biomarkers might be useful to predict which at-risk fetuses will go on to have FASD postnatally. 

### 4.8. Limitations and Future Directions

Several factors limit the generalizability of the present findings. Since the study is observational (e.g., women were not assigned randomly to EtOH-exposed vs. unexposed groups prospectively), associations of biomarker abnormalities with exposure to EtOH or other drugs can only be described in terms of correlation rather than causality. It also may be that there are differences in affective dysregulation and EtOH use among women who elect to terminate their pregnancy compared to those who do not. Moreover, the relatively limited sample size of this study has prevented us from controlling for all the variables that might be relevant, such as maternal smoking, obesity, race, or socioeconomic status. The next step is a much larger prospective study, for which IRB approval has already been obtained, of pregnancies that were not terminated, in which the children are followed postnatally to determine whether the promising FB-E biomarkers we have identified in the present and previous studies can predict which fetuses will go on to have FASD postnatally and which of these will include affective dysregulation. The results also might suggest strategies to prevent psychiatric disorders in children exposed to EtOH during their fetal development. 

## Figures and Tables

**Figure 1 cells-13-00002-f001:**
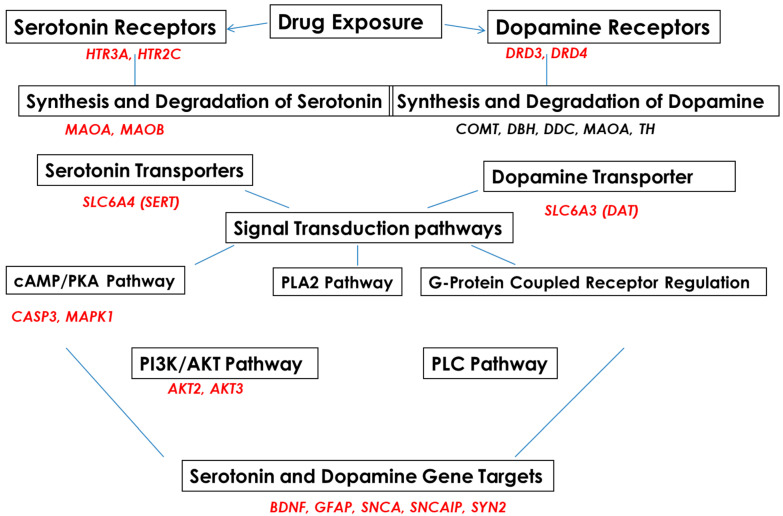
Serotonin and dopamine pathways and upstream and downstream target genes in fetal brains, studied with RNA arrays.

**Figure 2 cells-13-00002-f002:**
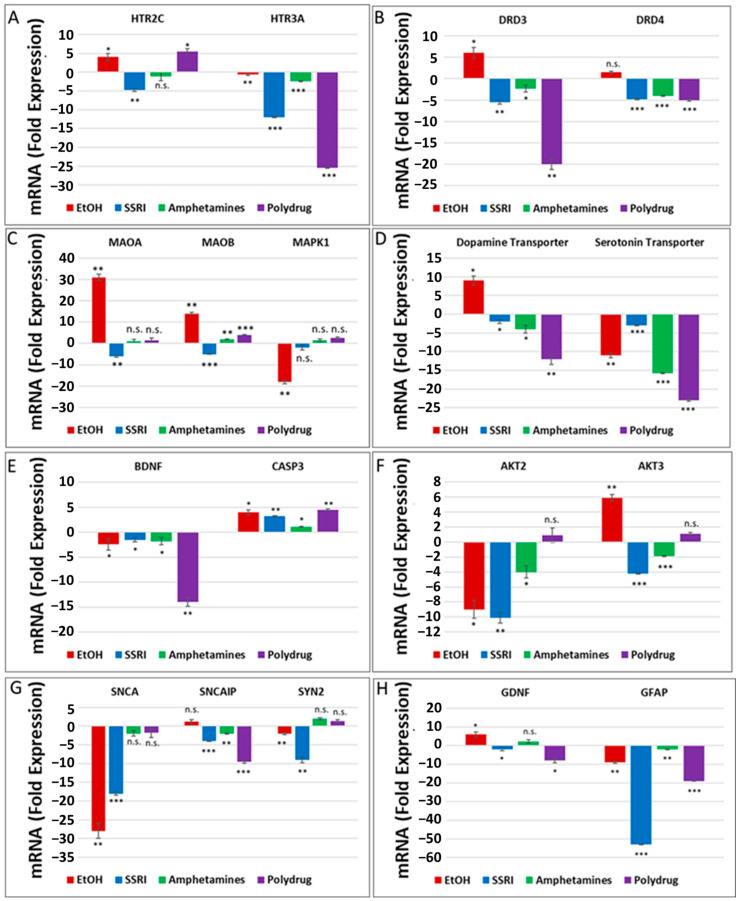
Prenatal EtOH, SSRI, and polydrug exposures disrupt 5-HT and DA pathways in fetal brain. Brains from fetuses with exposure to EtOH (n = 6), SSRIs (n = 5), EtOH + SSRIs (n = 3) amphetamines (n = 2) or polydrugs (n = 6) were compared with unexposed controls (n = 12), as enumerated in Table 1. The brains were assayed for expression of molecules in the 5-HT and DA pathways by qRT-PCR on micro-arrays. Downregulation for most genes was greatest in the cases with EtOH and SSRI exposure (graphs show means from triplicate assays +/− SD). The *p-*values shown in this figure and in Figure 3 are only for the comparison between the indicated exposure group and controls. In several cases, there are significant differences between exposure groups, and these are not shown here to avoid clutter but are commented on in the text. * *p* < 0.05, ** *p* < 0.01, and *** *p* < 0.001; n.s.: not significant. Values are shown in fold change in expression, normalized to the housekeeping gene for actin. (**A**). RNA levels for the 5-HT receptor HTR3A were downregulated in EtOH- and SSRI-exposed fetal brains. (**B**). Maternal EtOH use increased mRNA for DA receptor DRD3 in fetal brain, while SSRI and polydrug use downregulated DRD3. (**C**). Effect of EtOH and poly drug use on MAOA, MAOB, and MAPK1 in 5-HT and DA pathways in fetal brain. (**D**). EtOH exposure increased DAT and reduced SERT mRNAs in fetal brain. All other drug exposures showed downregulation of both. (**E**). All drug exposures inhibited mRNA expression of BDNF, which is downstream of the 5-HT/DA/NE pathways, and upregulated caspase-3 in fetal brain. (**F**). EtOH exposure downregulated RNA expression of AKT2 and upregulated AKT3 in fetal brain. SSRIs and amphetamines downregulated both, while polydrug exposure showed no significant effects. (**G**). Downregulation of RNA expression for some synaptic genes in fetal brain by exposure to EtOH, SSRI, and amphetamines. (**H**). SSRI and polydrug exposure inhibited mRNA expression of GDNF and GFAP, which are downstream genes in the 5-HT and DA pathways.

**Figure 3 cells-13-00002-f003:**
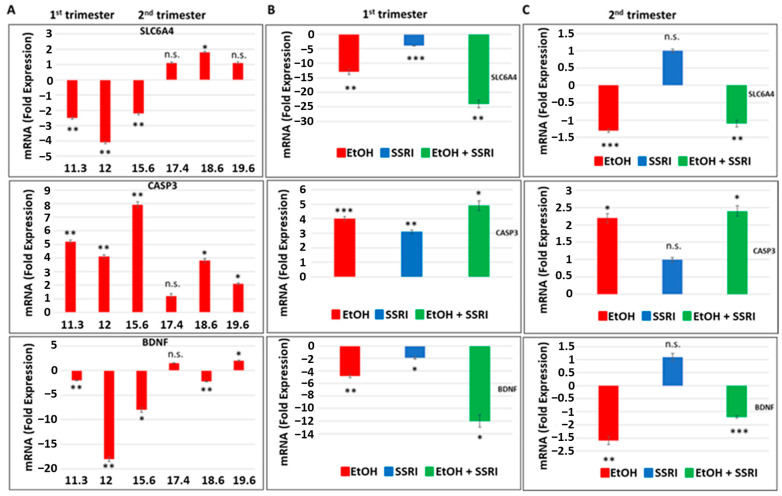
Effects of early or late exposure to EtOH and SSRI on gene expression for the 5-HT transporter SLC6A4, BDNF, and caspase-3. RNAs from the brain of fetuses with early (1st trimester) or late (2nd trimester) exposure to EtOH, SSRIs, or both, and from non-exposed controls, were studied for SLC6A4 (the gene coding for SERT), caspase-3 and BDNF gene expression by qRT-PCR. The numbers of fetuses studied were as in Table 1: EtOH-exposed (n = 6), SSRI (n = 5), EtOH + SSRI (n = 3) and unexposed controls (n = 12). Values for qRT-PCR are shown in fold change of expression normalized to actin. Error bars are averaged values for the means of triplicate assays +/− SD. * *p* < 0.05, ** *p* < 0.01, and *** *p* < 0.001. n.s.: not significant. (**A**). Early EtOH exposure was associated with downregulation of SLC6A4 (top), upregulation of caspase-3 (middle), and downregulation of BDNF (bottom). Late exposure showed mainly up-regulation of caspase-3, while SLC6A4 and BDNF showed only inconsistent or small changes. (**B**). Synergistic effects of early EtOH and SSRI exposure on mRNA expression of SLC6A4 (top), caspase-3 (middle panel), and BDNF (bottom). Early exposure to EtOH or SSRIs induced downregulation of SLC6A4 and BDNF, and upregulation of caspase-3. In each case, the effects of EtOH + SSRIs were greater than those of either exposure alone. (**C**). Similarly, late exposure to EtOH was associated with downregulation of SLC6A4 (top) and BDNF (bottom) genes, while caspase-3 was upregulated (middle), but the changes were less pronounced than in the 1st trimester. Moreover, the effects of late exposure to SSRIs produced no significant effects on these three markers, and the combination of EtOH + SSRI was not significantly greater than either alone (n = 12 controls, n = 6 EtOH, n = 3 SSRI, and n = 3 EtOH + SSRI).

**Figure 4 cells-13-00002-f004:**
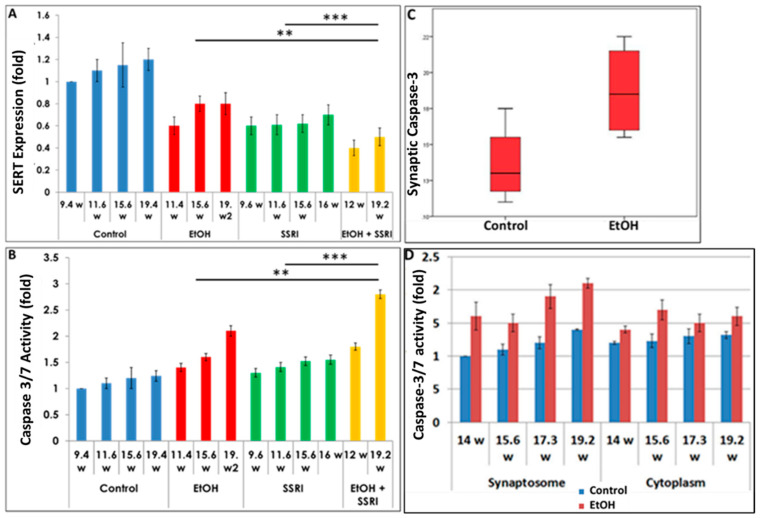
Effects of gestational age, EtOH, and SSRIs on fetal synaptosome SERT protein expression and on neuronal injury. Synaptic and cytoplasmic extracts were prepared and SERT levels were analyzed by qWestern blot assays using the anti-serotonin transporter antibody AB10514P (Millipore). Apoptosis was assessed in fetal brain tissue by analysis of caspase-3 activation, as described in Methods. (**A**). Exposure to either EtOH or SSRIs was associated with reduction in SERT levels in fetal brain synaptosomes. The effect of EtOH + SSRI was greater than that of either alone. (**B**). Activation of caspase-3 in the same cases. Exposure to either EtOH or SSRIs increased activated caspase-3 levels in fetal brain synaptosomes. The effect of EtOH + SSRI was significantly greater than that of either exposure alone. (**C**). EtOH exposure was associated with increased caspase-3 activation in fetal brain synaptosomes (n = 10 EtOH vs. 10 controls; *p* < 0.05). (**D**). In 4 EtOH-exposed fetal brains, caspase-3 activity was significantly increased (*p* < 0.05) in both synaptosomes and, to a lesser degree, in cytoplasm, compared to their GA- and sex-matched controls. Each bar represents the mean and SE of three determinations in one fetus. ** *p* < 0.01, and *** *p* < 0.001.

**Figure 5 cells-13-00002-f005:**
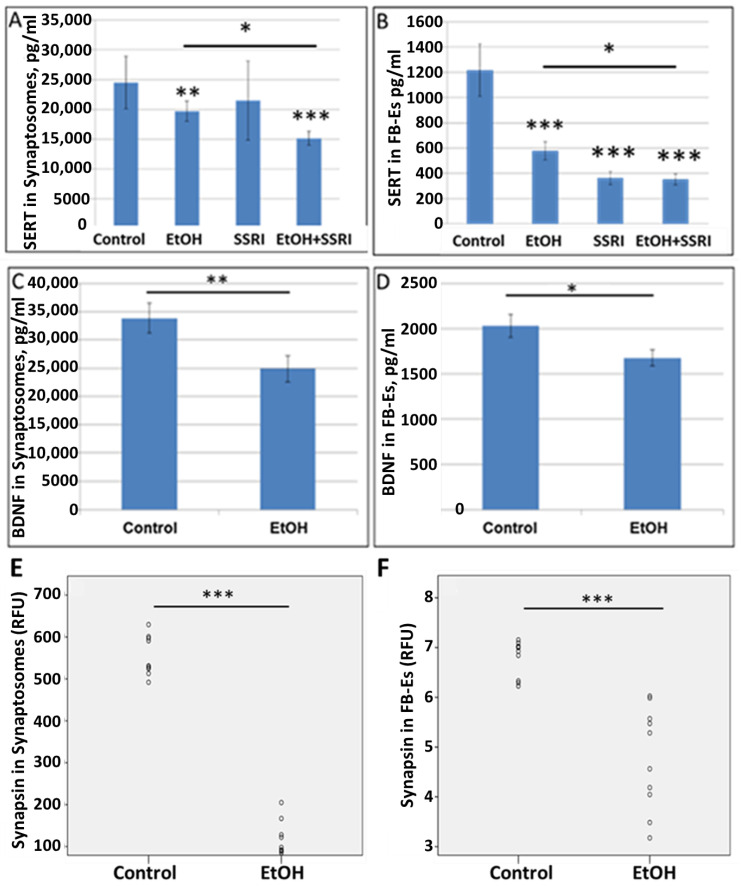
EtOH exposure inhibits expression of SERT and synaptic proteins in both synaptosomes and exosomes. (**A**). SERT and BDNF levels were quantified by ELISA and values expressed in pg/mL, after normalization to the exosomal marker protein CD81. EtOH and SSRI each reduced SERT levels (measured by ELISA; n = 6/group) in synaptosomes, and the effects were additive. (**B**). EtOH and SSRI each reduced SERT levels in FB-Es (ELISA n = 6/group), and the effect of combined EtOH + SSRI was about the same as that of SSRI alone. (**C**). Exposure to EtOH reduced BDNF levels in synaptosomes (ELISA; n = 10/group). (**D**). BDNF levels were reduced in FB-Es of EtOH-exposed cases (*p* < 0.05; n = 10/group). Graphs show means from triplicate assays +/− SD. (**E**). Exposure to EtOH reduced synapsin levels compared to controls (n = 10 cases/group) in synaptosomes and in exosomes (**F**), as assayed by quantitative Western blot, and shown in relative fluorescence units (RFU). * *p* < 0.05, ** *p* < 0.01, and *** *p* < 0.001.

**Figure 6 cells-13-00002-f006:**
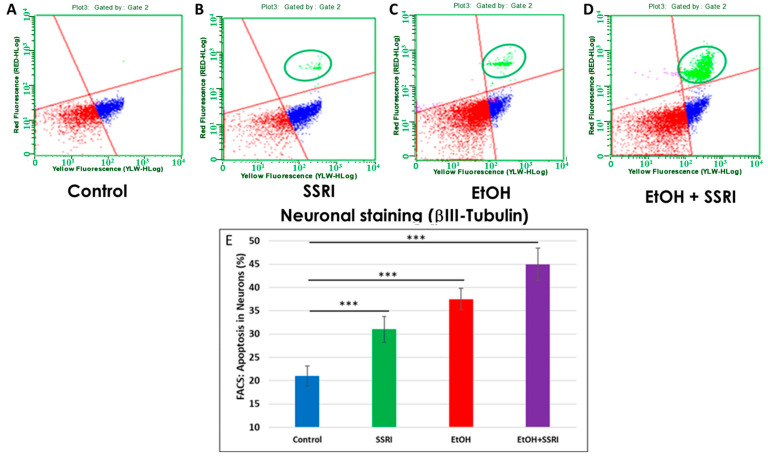
Neuronal apoptosis in EtOH- and SSRI-exposed fetal brains. Neuronal viability in human fetal brain was assessed by FACS analysis, using multiple fluorescent dyes to sort cell types. Representative dot-plots of the flow cytometry to measure membrane changes associated with apoptosis, using Annexin V-PE and 7-AAD to identify dead cells, together with the neuronal marker βIII-tubulin. Three neuronal populations are identified with the Guava Nexin Reagent and analyzed with the Guava Nexin Software (5.3.1): The lower left quadrant represents the live cells 7-AAD and annexin-V-PE negative (in red); the lower right quadrant represents cells with early signs of injury or enhanced vulnerability (apoptotic cells 7-AAD negative and Annexin-V-PE positive; in blue). The upper right quadrant represents cells with signs of progressive cell injury and cell death (necrotic cells 7-AAD and annexin-V-PE positive; in green). (**A**). Representative images from flow cytometry analysis in 6 control fetal brains in the late 1st through mid-2nd trimester. (**B**). Same as (**A**) for 5 fetuses exposed to SSRI. (**C**). Same as (**A**) for 6 fetuses exposed to EtOH. (**D**). Same as (**A**) for 3 fetuses exposed to both EtOH + SSRI. Late injury and cell death were detected in SSRI-exposed neuronal cells, more so in EtOH-exposed fetuses, and most in a combined EtOH + SSRI exposure. (**E**). Apoptosis in neurons (% of neuronal cells expressing activated caspase-3). Graphs represent average data from flow cytometry measures in control brains (n = 6), SSRI (n = 5), EtOH (n = 6), and EtOH + SSRI (n = 3). *** for *p* < 0.001.

**Figure 7 cells-13-00002-f007:**
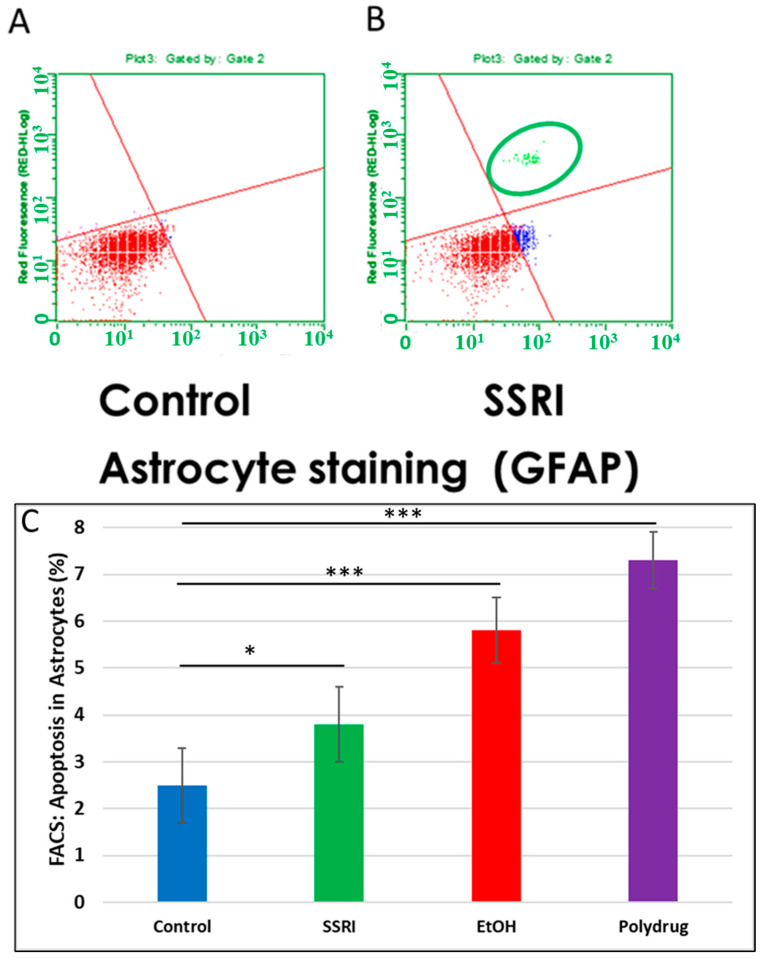
Increased apoptosis in astrocytes after SSRI exposure. GFAP was used as an astrocytic marker to assess glial viability by FACS analysis (n = 6 control and 5 SSRI-exposed fetal brains). Annexin V was used as a nexin assay for early apoptosis, and 7-AAD for late apoptosis. (**A**). No apoptotic cells were detected in control astrocytic cells (all cells in the left lower quadrant; red). (**B**). SSRI exposure increased apoptosis of astrocytic cells. As in Figure 6, the upper right quadrant represents cells with signs of progressive injury and death (green). The lower right quadrant represents cells with early signs of injury (blue). (**C**). Apoptosis (% of astrocytic cells expressing activated caspase-3). Graphs represent average data from flow cytometry measures in control brains (n = 6), SSRI (n = 5), EtOH (n = 6), and polydrug (n = 3). * *p* < 0.05, *** *p* < 0.001.

**Figure 8 cells-13-00002-f008:**
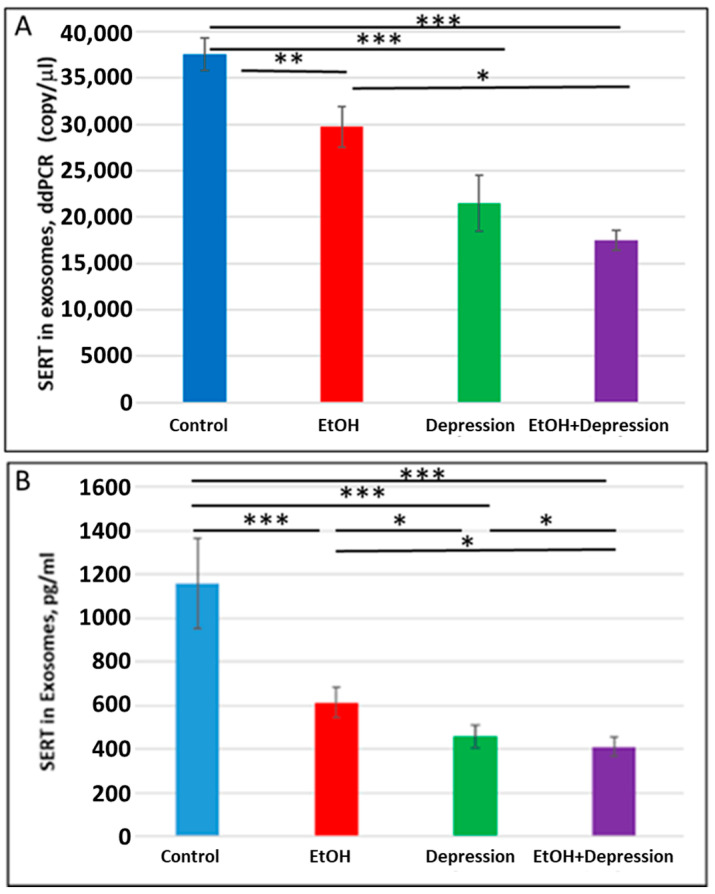
Maternal use of EtOH is associated with downregulation of SERT mRNA and protein in FB-Es. FB-Es isolated from the plasma of mothers who drank EtOH (n = 10) or did not drink EtOH (n = 10) and who did or did not suffer from depression (n = 10 each group) during pregnancy were isolated from maternal blood and assayed for SERT mRNA by ddPCR and for SERT protein by ELISA. For absolute quantitation of SERT mRNA in exosomes by ddPCR, values are shown in copies/μL. For quantitation of SERT protein by ELISA, values are shown in pg/mL (normalized to CD81). (**A**). EtOH and depression each were associated with downregulation of SERT mRNA levels in FB-Es. Downregulation was greatest in the cases with EtOH + depression, although the difference between the combined effect compared with that of depression alone was not statistically significant. (**B**). Downregulation of SERT protein levels in FB-Es by EtOH and depression. The same FB-E preparations were studied by ELISA for SERT protein levels. EtOH and depression each reduced SERT levels, and the effect of the combination of EtOH + depression was significantly stronger than that of EtOH exposure or depression alone. Graphs show means from triplicate assays +/− SD (n = 10 fetuses/group); all comparison differences were significant at * *p* < 0.05 and ** *p* < 0.01, *** *p* < 0.001, or less.

**Figure 9 cells-13-00002-f009:**
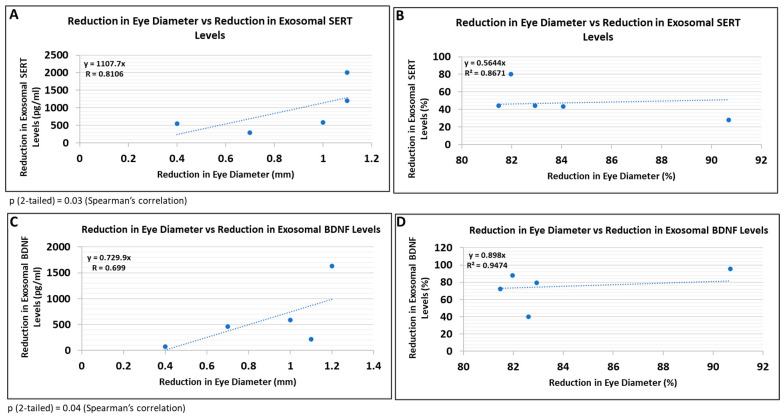
Reductions in FB-E SERT and BDNF levels correlate with reductions in eye diameter in fetuses exposed to EtOH + SSRIs. Eye diameters were measured in histological sections of human fetuses that had been exposed to both EtOH and SSRI. FB-E SERT and BDNF levels were measured by ELISA. Each of five EtOH + SSRI-exposed fetal eyes from 1st- and 2nd-trimester pregnancies was paired with a GA- and sex-matched unexposed control. Their matching maternal blood samples were obtained at the time of voluntary pregnancy termination. Assays were performed in triplicate on contents of FB-Es isolated from the maternal blood. Correlation between reduction in eye size (difference between EtOH- or SSRI-exposed fetus and its paired control) and reduction in exosomal SERT (**A**,**B**) and BDNF (**C**,**D**) levels is presented as scatter plots. Data in (**B**,**D**) are presented in %. Calculations are based on Spearman’s correlation on exact two-tailed probabilities critical and *p*-values for N > 2 <= 18.

**Table 1 cells-13-00002-t001:** Drug exposure groups for most experiments using RNA arrays for 5-HT and DA pathways.

Subjects	Drugs	No. of Subjects
Control	-	12
EtOH	EtOH	6
SSRI	SSRI	5
Amphetamine	Amphetamine	2
Polydrug	Polypharmacy group	6
	EtOH + SSRI	1
	EtOH + SSRI + amphetamine	1
	EtOH + SSRI + opioid + amphetamine	1
	EtOH + opioid + amphetamine	1
	Opioid + amphetamine	1
	Benzodiazepines + tobacco + amphetamine	1

**Table 2 cells-13-00002-t002:** Correlations between protein levels in FB-Es and synaptosomes. Spearman’s Rho (non-parametric) correlation coefficient and *p-*value are presented for synaptic proteins. EtOH reduces synaptophysin, REST, synaptopodin, BDNF, and synapsin-2 levels in fetal brain synaptosomes (n = 10 EtOH and 10 controls).

Marker	Rho	*p*-Value
Synaptophysin	0.61	0.004
REST	0.55	0.01
Synaptopodin	0.47	0.004
BDNF	0.56	0.01
Synapsin-2	0.86	0.001

## Data Availability

Data are contained within the article.

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
