# Peer review of "Biomarkers of Affective Dysregulation Associated with In Utero Exposure to EtOH"

_cells, 2023, doi:10.3390/cells13010002_

Round 1

Reviewer 1 Report

Comments and Suggestions for Authors

Overall, this was a well-written manuscript that describes associations between prenatal alcohol exposure and biomarkers of dysregulation.  I do have a number of concerns that I have itemized below.

1.  Avoid causal language when describing these findings as well as the existing human literature.  Even the title of the manuscript suggests causality which is inappropriate in a quasi-experiment.  Instead using language like "associations."

2. Provide specific hypotheses

3. Were women screened for ETOH before tissue was sample or were all available tissue samples analyzed and then women were questioned about alcohol use?

4. Were other substances measured and considered analytically?

5.  There are a number of limitations of this study (in addition to those described above) that need to be acknowledged.  These include sample size which impacts generalizability as well as the fact that all analysis was done on tissue as the result of elective abortions.  It is likely that there are differences in dysregulation and ETOH use among women who elect to abort as compared to those that do not.

Author Response

Overall, this was a well-written manuscript that describes associations between prenatal alcohol exposure and biomarkers of dysregulation.  I do have a number of concerns that I have itemized below.

  1. Avoid causal language when describing these findings as well as the existing human literature.  Even the title of the manuscript suggests causality which is inappropriate in a quasi-experiment.  Instead using language like "associations."

A: Although we used “association” in several places, in other places we did use language that implied causality.  We now have changed the title and, throughout the text, replaced all implications of causality with the more cautious “association”.

  1. Provide specific hypotheses

A: In the revised Abstract we point out that the mood disorders of FASD “…could conceivably be due to postnatal social or environmental factors. However, we postulate that more likely, the affective dysregulation is associated with effects of EtOH exposure on the development of fetal serotonergic (5-HT) and/or dopaminergic (DA) pathways, i.e., pathways that in postnatal life are believed to regulate mood.” Then we state: “We hypothesized that maternal use of EtOH and/or SSRIs during pregnancy would be associated with impaired fetal neural development, detectable as abnormal levels of monoaminergic and apoptotic biomarkers in FB-Es.

  1. Were women screened for ETOH before tissue was sample or were all available tissue samples analyzed and then women were questioned about alcohol use?

A:  Women first were screened for EtOH use, and then tissue was collected. We added this sentence to Methods on page 8.

  1. Were other substances measured and considered analytically?

A: Other substances (opioids, SSRIs, amphetamines) had already been measured for previous studies on placental transport activities (added on page 8).

  1. There are a number of limitations of this study (in addition to those described above) that need to be acknowledged.  These include sample size which impacts generalizability as well as the fact that all analysis was done on tissue as the result of elective abortions.  It is likely that there are differences in dysregulation and ETOH use among women who elect to abort as compared to those that do not.

A: We have added the following section to the end of Discussion: “Limitations and Future Directions.  Several factors limit the generalizability of the present findings.  Since the study is observational (e.g., women were not assigned randomly to EtOH-exposed vs. unexposed groups prospectively), associations of biomarker abnormalities with exposure to EtOH or other drugs can only be described in terms of correlation rather than causality. It also may be that there are differences in affective dysregulation and EtOH use among women who elect to terminate their pregnancy compared to those that do not.  Moreover, the relatively limited sample size of this study has prevented us from controlling for all the variables that might be relevant, such as maternal smoking, obesity, race or socioeconomic status. The next step is a much larger prospective study, for which IRB approval has already been obtained, of pregnancies that were not terminated, in which the children are followed postnatally to determine whether the promising FB-E biomarkers we have identified in the present and previous studies can predict which fetuses will go on to have FASD postnatally, and which of these will include affective dysregulation. The results also might suggest strategies to prevent psychiatric disorders in children exposed to EtOH during their fetal development.”

Reviewer 2 Report

Comments and Suggestions for Authors

The manuscript is exciting as it sheds new light on the developmental processes of the fetal brain and drug, alcohol, and SSRI use effects during pregnancy. The study's implications are significant for understanding the biological mechanisms that can help treat ETOH and drug toxicity. However, I suggest some minor changes that may improve the clarity of the manuscript:

1. A glossary of acronyms can be included at the beginning of the manuscript to facilitate text comprehension.
2. increasing the sample size can be challenging, but adding a curly bracket that includes the "polypharmacy group" case in parentheses can improve Table 1.

3. A note explaining the limited number of cases should be added to highlight the need for further studies.

4. Including a better description of the brain tissues examined in the various tests may be helpful. This could improve our understanding of receptor interactions.

Author Response

The manuscript is exciting as it sheds new light on the developmental processes of the fetal brain and drug, alcohol, and SSRI use effects during pregnancy. The study's implications are significant for understanding the biological mechanisms that can help treat ETOH and drug toxicity. However, I suggest some minor changes that may improve the clarity of the manuscript:

1. A glossary of acronyms can be included at the beginning of the manuscript to facilitate text comprehension.

A: We have moved the list of Abbreviations to the beginning of the text (page 3).

  1. increasing the sample size can be challenging, but adding a curly bracket that includes the "polypharmacy group" case in parentheses can improve Table 1.

    A: The table has been reformatted as suggested.

  1. A note explaining the limited number of cases should be added to highlight the need for further studies.

A: A new paragraph has been added to Discussion that describes limitations, including sample size (page 19), see above. Moreover, in Materials and Methods, the limitation of sample sizes is further clarified: “The selection of cases and controls was based on the availability of intact fetal brain tissues, matching maternal blood samples, and the availability of data for matching of sex, and GA (Darbinian et al, 2021, 2023ab, Goetzl et al, 2016, 2019ab). Thus, each EtOH- and/or drug-exposed fetus involved in a comparison was individually paired with a GA- and sex-matched control.”

  1. Including a better description of the brain tissues examined in the various tests may be helpful. This could improve our understanding of receptor interactions.

A: On page 9, we have added: “Initial histologic staining of brain tissues from the Biobank confirmed that we had collected mostly cerebral cortex.”

Reviewer 3 Report

Comments and Suggestions for Authors

The manuscript by Darbinian and colleagues entitled “Effects of in utero EtOH exposure on biomarkers of affective dysregulation” aims to quantify changes in a variety of molecular markers in blood and fetal tissue following exposure to ethanol, SSRIs, or a combination of ethanol and other drugs (polydrug). Specifically, the dopamine and serotonin systems and associated downstream signals were targeted by their analyses. Overall, the authors found a variety of alterations in their analyses suggesting that prenatal ethanol or SSRI exposure have similar effects on some systems, and that polydrug exposure seems to magnify the observed alterations. This study is significant as few studies have used human tissue to understand the molecular alterations resulting from prenatal drug exposure as a means to discover potential biomarkers. However, the study was difficult to read, with a lot of facts provided that were irrelevant to the primary focus of the study. Additionally, there are several concerns that need to be addressed and considered when interpreting the findings. Below are suggestions that should be taken into consideration:

1)    In figure 3, it is unclear what the numbers on the x-axis of panels A signify? Are those the gestational weeks when the samples were collected?

2)    Regarding figure 4D, there were samples from control and ethanol exposed that matched exactly for gestational week (i.e. 14 w, 15.6 w, etc.)?

3)    It's unclear what is the functional difference between synaptic versus cytosolic caspase expression since caspases are typically found in the cell body. This should be explained.

4)    The resolution of the y-axis in Figure 5E and F is illegible.

5)    There are no statistical analyses for the flow cytometry data. Although it is clear from the graphs that there are more green dots in the EtOH group versus SSRI, and EtOH + SSRI (which should be called polydrug) versus the others, this is simply a qualitative observation.

6)    If the EtOH group also increases caspase activity in astrocytes, the graph should be included in Figure 7 as it is for figure 6. Additionally, the graph for the polydrug group should also be included even if there weren't any changes.

7)    Given that the fetuses used in this study varied across gestational age, it is concerning that eye diameter was used as a variable since the eye diameter likely changes quite a bit from 10-20 weeks of gestation.

8)    The intro and discussion are very choppy and difficult to read. It is written as a list of facts that are often disconnected. Additionally, it is unclear why there is so much discussion about the role of the DA and 5-HT systems in alcohol use disorder and as targets to treat alcohol use disorder when the purpose of the study was to measure changes in some of these systems following prenatal exposure to alcohol and other drugs.

Comments on the Quality of English Language

There are some minor grammatical and syntax errors throughout the document.

Author Response

Reviewer 3

The manuscript by Darbinian and colleagues entitled “Effects of in utero EtOH exposure on biomarkers of affective dysregulation” aims to quantify changes in a variety of molecular markers in blood and fetal tissue following exposure to ethanol, SSRIs, or a combination of ethanol and other drugs (polydrug). Specifically, the dopamine and serotonin systems and associated downstream signals were targeted by their analyses. Overall, the authors found a variety of alterations in their analyses suggesting that prenatal ethanol or SSRI exposure have similar effects on some systems, and that polydrug exposure seems to magnify the observed alterations. This study is significant as few studies have used human tissue to understand the molecular alterations resulting from prenatal drug exposure as a means to discover potential biomarkers. However, the study was difficult to read, with a lot of facts provided that were irrelevant to the primary focus of the study. Additionally, there are several concerns that need to be addressed and considered when interpreting the findings. Below are suggestions that should be taken into consideration:

1)    In figure 3, it is unclear what the numbers on the x-axis of panels A signify? Are those the gestational weeks when the samples were collected?

A: Yes, the numbers on the x-axis represent gestational weeks. We added this explanation to the figure legend.

2)    Regarding figure 4D, there were samples from control and ethanol exposed that matched exactly for gestational week (i.e. 14 w, 15.6 w, etc.)?

A: Yes, as we explained in Materials and Methods, each EtOH- or drug-exposed fetus was matched with a GA- and fetal sex-matched control. This has now been clarified: “The selection of cases and controls was based on the availability of intact fetal brain tissues, matching maternal blood samples, and the availability of data for matching of sex, and GA (Darbinian et al, 2021, 2023ab, Goetzl et al, 2016, 2019ab). Thus, each EtOH- and/or drug-exposed fetus involved in a comparison was individually paired with a GA- and sex-matched control.”

3)    It's unclear what is the functional difference between synaptic versus cytosolic caspase expression since caspases are typically found in the cell body. This should be explained.

A: We have added the following section in Discussion: “Increased levels of activated caspase-3 in synaptosomes.  Transient non-apoptotic activation of caspases has been described in dendritic pruning during normal development (Kuo et al, 2006; Ertuk et al., 2014) and some other cell functions.  Local, non-apoptotic caspase-3 activation is involved in dendritic spine loss and synaptic dysfunction in Alzheimer’s disease, and in the rapid loss of dendritic spines seen with synaptic long-term depression (LTD) in striatal projection neurons forming the indirect pathway. Systemic treatment with a caspase inhibitor prevented both the dendritic spine pruning and the physiological deficit of LTD without interfering with the ongoing dopaminergic degeneration (discussed in (Fieblinger et al, 2022)).  In the present study, the increased levels of activated caspase-3 demonstrated in synaptosomes from EtOH-exposed fetuses is consistent with the known loss of brain volume and the cognitive disabilities of children with FAS (Pfefferbaum et al., 2023) and the loss of dendritic spines previously shown in animal fetuses exposed to EtOH (Cui et al, 2010, Clabough et al, 2021) and in human FAS (Ferrer et al, 1987).  Whether this involves monoaminergic neurons remains to be determined.

4)    The resolution of the y-axis in Figure 5E and F is illegible.

A: Figures 5E and 5F have been replaced.

5)    There are no statistical analyses for the flow cytometry data. Although it is clear from the graphs that there are more green dots in the EtOH group versus SSRI, and EtOH + SSRI (which should be called polydrug) versus the others, this is simply a qualitative observation.

A: Statistical analyses for flow cytometry are added in the figures.

6)    If the EtOH group also increases caspase activity in astrocytes, the graph should be included in Figure 7 as it is for figure 6. Additionally, the graph for the polydrug group should also be included even if there weren't any changes.

A: Graphs have been added as requested.

7)    Given that the fetuses used in this study varied across gestational age, it is concerning that eye diameter was used as a variable since the eye diameter likely changes quite a bit from 10-20 weeks of gestation.

A: What we graphed is not the eye diameters, but the reduction of eye diameters in exposed fetuses compared to their age-matched controls.  Since eye diameter is one of the facial hallmarks of FASD, we correlated biomarkers with eye diameter as a potential way to indicate whether a fetus already showed evidence that it might have developed FASD postnatally. To measure eye diameter as accurately as possible, we used a digital caliper. Because eye diameter increases with GA, the absolute difference between exposed and unexposed controls increases, as shown in the graphs. We now add graphs of percent change in eye diameter (compared to the age-matched control) vs. percent change in biomarkers, which show a relatively constant ratio, i.e., the graphs have slopes close to 0.

8)    Comments on the Quality of English Language. There are some minor grammatical and syntax errors throughout the document. The intro and discussion are very choppy and difficult to read. It is written as a list of facts that are often disconnected. Additionally, it is unclear why there is so much discussion about the role of the DA and 5-HT systems in alcohol use disorder and as targets to treat alcohol use disorder when the purpose of the study was to measure changes in some of these systems following prenatal exposure to alcohol and other drugs.

A: The text has been edited to remove some redundancies, focus more on the main topic and eliminate remaining grammatical and syntactical errors.  The goal of this paper is to determine whether EtOH exposure in utero is associated with changes in 5-HT and DA pathways that are implicated in mood disorders in adults.  The ultimate goal is to be able to predict which at risk fetuses would go on to have FASD postnatally and which of these would show affective dysregulation.  We hope that the changes described above make this clearer. 

Round 2

Reviewer 1 Report

Comments and Suggestions for Authors

Thank you for your thoughtful and appropriate responses to the previous set of reviews.  I have no additional concerns.